# An Integrated Spatial Autoregressive Model for Analyzing and Simulating Urban Spatial Growth in a Garden City, China

**DOI:** 10.3390/ijerph191811732

**Published:** 2022-09-17

**Authors:** Bingkui Qiu, Min Zhou, Yang Qiu, Shuhan Liu, Guoliang Ou, Chaonan Ma, Jiating Tu, Siqi Li

**Affiliations:** 1Department of Tourism Management, Jin Zhong University, Jinzhong 033619, China; 2College of Public Administration, Huazhong University of Science and Technology, Wuhan 430074, China; 3Department of Economics, University College of London, London WC1E 6BT, UK; 4School of Construction and Environmental Engineering, Shenzhen Polytechnic, Shenzhen 518055, China

**Keywords:** urban spatial growth simulation, GIS, cellular automata, spatial autoregressive, Chengdu

## Abstract

In the past, the research on models related to urban land-use change and prediction was greatly complicated by the high precision of models. When planning some garden cities, we should explore a more applicable, specific, and effective macro approach than the community-level one. In this study, a model consisting of spatial autoregressive (SAR), cellular automata (CA), and Markov chains is constructed. One It can well-consider the spatial autocorrelation and integrate the advantages of CA into a geographical simulation to find the driving forces behind the expansion of a garden city. This framework has been applied to the urban planning and development of Chengdu, China. The research results show that the application of the SAR model shows the development trend in the southeast region and the needs to optimize the central region and protect the western region as an ecological reserve. The descriptive statistics and the spatial autocorrelation of the residuals are reliable. The influence of spatial variables from strong to weak is distance to water, slope, population density, GDP, distance to main roads, distance to railways, and distance to the center of the county (district). Taking 2005 as the initial year, the land-use situation in 2015 was simulated and compared with the actual land-use situation. It seems that the Kappa coefficient of the construction-land simulation is 0.7634, with high accuracy. Therefore, the land use in 2025 and 2035 is further simulated, which provides a reference for garden cities to formulate a reasonable urban space development strategy.

## 1. Introduction

Technological progress has promoted the development of modern society. In pursuit of the single objective of the economy, land-resource utilization activities in developing countries have also been seriously fragmented [1,2,3]. Over- and irrational utilization like farmland loss, soil quality degradation, biodiversity reduction, and so on exist in land-resource utilization, triggering natural resource depletion and ecological degradation. Therefore, conservation and restoration strategies are particularly important in land-use planning. A garden city was proposed by Ebenezer Howard (1902), who stated that a rural strip should be consistently preserved around the city as the proper principle of urban development for providing ecological services and a convenient life, and for the protection of the natural environment. For instance, vegetable gardens in communities and vegetable plots around the cities are not only an important symbol of the rural–urban fringe but also the key to solving the problem of urban and rural environmental cognitive dissociation [4,5]. Furthermore, developed countries and cities, such as Germany [6], Los Angeles in the United States [7], and Welwyn in the United Kingdom [8], are committed to both the establishment of a garden city and the ecology of a community. A garden city is often designed and managed at the community level, which is a typical micro land-use behavior. So the spatial development of a garden city should also be explored through the “bottom-up” approach [9], but many existing methods are limited to specific cases, having no universal applicability. Thus, the models related to urban land-use change and prediction should be relatively simplified as far as possible, rather than blindly pursuing complexity. In due course, a more macro-level approach that is more applicable, adaptable to more unique cases, and effective than the community-level one should be explored.

Land, an important material basis and space for human survival and development and an intricate complex of natural, historical, and economic attributes, develops under the dual influence of nature and society. Therefore, land-use change of each region happens all the time. Methods like field-location surveys and remote-sensing monitoring can be used to search for information about land-use change, but both of them can only be applied to investigate the changes that have occurred. However, the future direction and situation of land development need to be predicted based on simulation methods, which are often used to obtain observations of future land use to guide decision-making and improve land policies [10]. Urban land is the carrier of social and economic activities, natural and artificial landscapes, and so on. Dynamic urban change, whether planned or unplanned, will change the structure, shape, and function of built-up and non-built-up areas [11]. Analysis of urban land-use change can make urban sustainability deeply understood and developed. Moreover, the construction of new-type urbanization needs urban decision-makers to stand in a pre-emptive position and think from a long-term and humane perspective. Therefore, this study constructs a model that consists of spatial autoregressive (SAR), cellular automata (CA), and Markov chains. It can consider spatial autocorrelation well and integrate the geographical advantages of CA simulation to help to seek the driving forces behind the expansion of a garden city, which could provide a certain reference for the formulation of reasonable urban spatial development strategies for a garden city.

An urban land system is characterized by complexity, dynamics, and uncertainty. First, the land system is complex, for its development process involves physics, biology, chemistry, social science, and other fields, and covers multiple theories. Its complexity is a whole caused by local interactions, which can be described by the “bottom-up” approaches, such as cellular automata (CA) and multi-agent, and the integrated models based on them. Second, the development of land management and planning requires a deep understanding of the dynamics of land systems. With the application of systematic science, land-change models have also begun to be adapted to have complex dynamic characteristics. In addition, the subjective uncertainty of land utilization and the objective uncertainty of natural environment change, social environmental turbulence, economic fluctuation, and scientific and technological progress will emerge constantly. Thus, it can be seen that the complexity, dynamics, and uncertainty of a land system are not only related to natural attributes but also closely bound up with human activities. The characteristics of a land system determine the difficulty of modeling.

Urban expansion problems are well dealt with by a scenario simulation model, which can simulate land-use situations under different development scenarios based on historical land-use data, and then put forward the scheme of how to optimize the quantity and spatial allocation of land resources under certain circumstances [12], such as multi-agent, system dynamics (SD), CLUE-S, Bayesian network, artificial neural networks (ANN), and CA.

The multi-agent model is widely used in land-use simulation, which has made an outstanding contribution to the understanding of complex system theory and the decision-making behaviors of social agents [13]. Although there are conducts subject to the land-unit scale and the principle of agent decision-making [14,15], the multi-agent model is not generalizable because of the lack of a unified spatial scale. Moreover, the definition and decision-making of agents are difficult to determine. Therefore, in terms of application, the multi-agent model has limitations, whether it is used alone or integrated with other models.

System dynamics (SD) is widely used in decision-making support, mainly in the area of energy and resource environmental protection [16,17]. For instance, Glöser et al. [18] used SD to study global copper flows. Zuser and Rechberger [19] employed SD to explore the scarcity of photovoltaic resources, and others such as Rui et al. [20], Swinerd and McNaught [21], Alejandro Ruiz Galeano and Cecilia Bautista Rodríguez [22], and Yao et al. [23]. In terms of stakeholders and strategies, however, the effect of SD remains limited and it lacks “spatial embodiment” with poor adaptive ability. Compared with SD, CLUE-S focuses on local interpretation and finds it difficult to represent a macro interaction process. For example, Peng et al. [24] used the CLUE-S model to simulate the dynamics of wetlands in cities. Wu et al. [25] applied the CLUE-S model to land-use simulation in Jiangsu Province, China. Shi et al. [26] determined wetland degradation trajectory by the CLUE-S model. Others like Lamichhane and Shakya [27], Li and Song [28], Tang et al. [29], and Tian et al. [30] also used it. Different from the above two methods, the Bayesian network is suitable for dealing with uncertainty problems, but it copes with huge calculations with difficulty [31,32], while ANN is expert at solving problems but is not good at dealing with uncertainty problems, and there are some deficiencies in the explanation of the driving forces [33,34].

As another method of understanding the complexity of a system [35], the “bottom-up” characteristic of the cell automata (CA) reflects the scenario that the local individual behaviors in a complex system constitute a global pattern. CA, very powerful in spatial calculation, is superior to other models in its effectiveness in reflecting the complex characteristics of land-use evolution. Thus, CA is suitable for the simulation of geographic processes [36]. In 1970, Tobler first used CA to simulate urban sprawl in Detroit, USA, and emphasized that the core feature of the geographic model is the neighborhood impact [37,38]. After that, CA has been widely applied to simulate various geographic processes, especially urban sprawl and land-use change, specifically involving the definition of transition rules, neighborhood impact, effect of spatial scale, model evaluation, precision detection, and case application [39,40]. More than that, CA is excellent at integrating with other models [41]. Rahman et al. [42] used the CA-ANN model to take, as an example, Asassuni Upazila and to explore the future land-use situation of the area for land-use planning. Omrani et al. [43] adopted the ANN-CA-based future land-use simulation (FLUS) model for a scenario-based flood-risk assessment. Tayyebi et al. [44] presented the CA model based on the distance dependence method (DDM) and the distance independent method (DIM), and the urban growth boundary (UGBs) of the metropolitan area of Tehran, Iran, was predicted with a superior fit. Tang [29] utilized the comprehensive evaluation and trade-off of the ecosystem service function (InVEST) model and integrated the CA model to study the influence of farmland expansion on carbon reserves under different scenarios in Hubei Province. Fu et al. [45] employed logistic regression united with CA to forecast and analyze land-use change.

As mentioned above, multi-agent, SD, ANN, CLUE-S, and CA have a commonality. When simulating the geographic process, they only take into account the impact of the neighborhood, whose transition rules overlook the high autocorrelation between land use and spatial variables [46,47]. In other words, there may be spatial clustering, non-negligible, or randomly distributed residual errors in the simulation process, which will affect the accuracy of the simulation results [35,48]. The first law of geography shows that the spatial entities are correlated by their distance: the closer the distance, the higher the correlation. On the contrary, the farther the distance, the lower the correlation [37,49]. Feng [35] compared logistic regression with spatial autoregression to find which one is better at taking into account spatial autocorrelation, and the latter performed well under the same conditions. Therefore, the spatial autoregression model can take good account of spatial autocorrelation [50].

Based on the above discussion, it is found that many existing methods are limited to specific cases, having no general applicability. So, this study constructs a model that consists of spatial autoregressive (SAR), cellular automata (CA), and Markov chains. It can consider spatial autocorrelation well and integrate the geographical advantages of CA simulation to seek the driving forces behind the expansion of a garden city. On the one hand, the approach that is more macroscopic can be effectively adapted to more unique cases; on the other hand, the present model can be applied to land-use planning practices to support decision-makers in developing appropriate policies for land-use planning.

## 2. Study Area

Chengdu (102°54′~104°53′ E, 30°05′~31°26′ N), an important western development platform in the Yangtze River Economic Belt, is located at the upper reaches of the Yangtze River and the eastern part of the Qinghai-Tibet Plateau. The highest elevation is 5364 m, the lowest 359 m, and the elevation of the heartland of Chengdu is about 750 m. Chengdu has a total area of 12,080 square kilometers and a resident population of more than 16 million. The proportion of primary, secondary, and tertiary industry was 3.5:43.7:52.8 (2015) [51]. As of 2015, Chengdu’s urban agglomeration included five central areas (Wuhou, Qingyang, Jinniu, Jinjiang, and Chenghua), other surrounding areas (Shuangliu, Longquanyi, Qingbaijiang, Xindu and Wenjiang, and Pidu), four counties (Jintang, Dayi, Xinjin, and Pujiang) and four county-level cities (Dujiang, Pengzhou, Chongzhou, and Qionglai). The administrative division map is shown in Figure 1. Relying on the good ecological environment and superior location of western China, Chengdu, the city with the world’s longest urban central axis in a single-center radiation structure, has become the most critical and fastest growing strategic highland in western China. Moreover, as the first garden city in China, Chengdu is also a new-type urbanization pilot city integrating internal optimization with external expansion. Thus, the proposed modeling framework can be adopted in the area.

The data in this paper are from the remote sensing monitoring database of the current land-use database by the Institute of Geographic Sciences and Natural Resources Research in China. The original data are processed to obtain other data, such as land-use data (30 m, www.resdc.cn, accessed on 16 September 2022), altimetric data and slope data (30 m, www.gscloud.cn, accessed on 16 September 2022), population density data, and GDP data (1 km, www.resdc.cn, accessed on 16 September 2022); it also obtains data like the distance to water, major roads, railways, and the center of the county (www.openstreetmap.org/, accessed on 16 September 2022) in 2005, 2010, and 2015. Figure 2 and Figure 3 show the land-use map of Chengdu in 2005 and 2015, respectively, and Figure 4 shows the elevation map of Chengdu. The land-use transition process is influenced by multiple factors such as physical geographical attributes, and socio-economic and restrictive conditions [52]. Physical geographical attributes refer to the elevation, slope, geology, soil, and so on. Social-economic conditions refer to the economy, population, policies, and so on. Restrictions include basic farmland, ecological protection areas, special land, control of the total amount of construction land, and so on [53]. Other important urban development driving forces also include distance to water, the center of the county (district), major roads, and so on. Among them, the roads network is an important part of the urban form, representing transportation costs, transportation convenience, economic accessibility, and the matching urban scale. In short, specific situations of the study area, such as slope (*Slope*), population density (*Population*), GDP (*GDP*), distance to the center of the county (*D_center_*), water (*D_water_*), major roads (*D_roads_*), and railways (*D_railways_*), are standardized and used for urban-space simulation in Chengdu. Then the random sampling method is used to extract sample points to train the transition rules.

## 3. Methodology

### 3.1. Markov Chain

This sub-section focuses on the formula of the Markov chain. The mathematical expression for the initial transition probability matrix $P_{m}$ is:(1)$$P_{m}=P_{IJ}=\left[\begin{array}{cccc} P_{11} & P_{12} & L & P_{1N}\\ P_{21} & P_{22} & L & P_{2N}\\ L & L & L & L\\ P_{N1} & P_{N2} & L & P_{NN} \end{array}\right]$$
where *N* is the number of states; $P_{IJ}$ represents the probability of transitioning from the state *I* to state *J*; and P_IJ needs to satisfy two conditions: ① $0\leq P_{IJ}\leq 1$, ② $\sum P_{IJ}=1\;\left(I,\;J=1,\;2,\;3,\ldots ,\;N\right)$.

According to the Markov model and conditional probability, the state vector *P*(*N* + 1) of the system at *N* + 1 time can be determined by its state vector *P*(*N*) and transition probability $P_{IJ}$ at *N* time [54]: $P\left(N+1\right)=P\left(N\right)\;PP_{IJ}$.

For *N*-order (*N*
≤ 2) transition probability, $P_{IJ}\;\left(N\right)$ is the probability of a random process from state E_I to state E_K through *M*(1≤M<N) times transition, and then from state $E_{K}$ to state $E_{K}$. Since there is no aftereffect, these transitions can also be regarded as independent from each other. So, the process can be expressed by the total probability formula:(2)$$P_{IJ}\;\left(N\right)=\sum P\_IK\;\left(M\right)\times P\_KJ\;\left(N-M\right)(1\leq M<N)$$

It can be seen from Equation (2) that the high-order transition probability matrix P_IJ(*N*) should be acquired by step-by-step matrix multiplication according to Equation (1).

### 3.2. Neighborhood Influence

CA determines the state change of a cell through the transition rules. The state of a cell at *t*+1 time is determined by the state of the time *t* and its neighborhoods [55]:(3)S_(t+1)=f(S_t,N)
where *S* refers to all possible state sets; *f* refers to transition rules; *N* is the neighborhood of a cell; and a square neighborhood is mostly used, such as the Moore neighborhood of *n* × *n*. In this paper, the default Moore (5 × 5) is selected as the neighborhood of a cell, and its kernel is as follows:$$\begin{array}{ccccc} 0 & 0 & 1 & 0 & 0\\ 0 & 1 & 1 & 1 & 0\\ 1 & 1 & 1 & 1 & 1\\ 0 & 1 & 1 & 1 & 0\\ 0 & 0 & 1 & 0 & 0 \end{array}$$

### 3.3. Spatial Autoregressive Model

The probability P_r of each non-built-up cell transitioning into a built-up cell can be fitted by the spatial autoregressive (SAR) model:(4a)Pr=e^(Yi)/(1+e^(Yi))
(4b)$$Y\_i=aX+\lambda wY_{i}+\theta$$
(4c)$$\theta =\mu Z+\rho ,\rho ∼N\left(0,\beta^{2}\right)$$
where $Y_{i}$ represents the influence of spatial variables on land-use transition; *a* is the parameter vector of spatial variable *X*; *λ* is the coefficient of the spatial lag variable $wY_{i}$; *w* is the spatial weight matrix adopting to the first-order Queen contiguity; *μ* is the autoregressive coefficient; *Z* is the spatial autoregressive structure; *θ* is the residual vector; and *ρ* satisfies the normal distribution [49]. When λ≠0 and *μ* = 0, it is a spatial lag model. It is not only related to the spatially independent variables but also takes into account the correlation between the spatially dependent variables. When *λ* = 0 and *μ* ≠ 0, it is a spatial error model, which implies that the model is more accurate.

### 3.4. Comprehensive Transition Probability

Comprehensive transition probability can be obtained by multi-criteria evaluation (MCE). The probability of any cell changing from a non-built-up state to a built-up state is denoted as P_a. This probability value is usually limited by the size of the spatial distance variables, such as distance to the center of the county (district), major roads, railways, and water [56].
(5)P_a=P_m×S×P_r×Res
where P_m refers to the transition probability from one state to another; *S* is the neighborhood influence; P_r is the transition probability of land use determined by spatial variables; and *Res* represents the restrictions including the large area of water, basic farmland, ecological reserves, and national parks that cannot be transitioned into built-up cells because of geographical constraints and protective policies [57].

### 3.5. Methods of Model Evaluation

In this paper, descriptive statistics and spatial autocorrelation of residuals are used to evaluate the goodness of the model [58]. Good models are those that have the average value of residuals close to 0, the sum of squares of residuals as small as possible, and the residuals randomly distributed. Therefore, the *p*-value should be large and if the *p*-value is greater than 0.05, the probability of random distribution is greater. The specific value and spatial autocorrelation statistics of residuals can be obtained through GeoD and GIS. The kappa index can be used as the main evaluation method for comparing the simulated land-use image with the actual land-use situation according to the division of the consistency of the kappa index range proposed by Landis and Koch [55]. When K≤ 0, there is no consistency; when 0 < K ≤ 0.2, it has a very low consistency; when 0.2 < K ≤ 0.4, it has a general consistency; when 0.4 < K ≤ 0.6, it has a medium consistency; when 0.6 < K ≤ 0.8, it has high consistency; and when 0.8 < K ≤ 1, it is almost completely consistent.

This study selects Chengdu, China, a garden city with the target of developing new-type urbanization, as the study area. Based on the current situation and deficiency of urban land-use structure and spatial arrangement, as well as regional development strategies, this study aims to achieve the following objectives: (1) obtaining the transition rules of land-use change by SAR fitting technology and evaluating the goodness of the fit. This work can deepen the excavation and enhance the learning of land systems. (2) The transition probability matrix is acquired by a Markov chain. Integrating the results of the Markov chain and SAR fitting into CA, the land-use suitability map is provided with pixel-level arrangement. (3) The spatial land use pattern of Chengdu in 2025 and 2035 are further obtained with the accuracy allowed. The framework of the optimized simulation process is shown in Figure 5. Meanwhile, China is in the process of conducting supply-side structural reform. In 2019, the central committee of the communist party of China and the state council issued a number of opinions on establishing a national territory development-plan system and supervising its implementation. There is a lot of discussion about it in China, and the results of this study can provide theoretical support for its formulation and raise awareness of optimizing urban-space structures.

## 4. Results

### 4.1. Transition Rules

As shown in Table 1, the fitting performance of the spatial variables shows that the average value of the residuals under the SAR model is −4 × 10^−8^, which is close to 0, revealing the reasonable residual distribution and fitting effect. The residuals’ Moran’s I index is 0.0225 and the *p*-value is 0.3087, making it clear that the residuals’ spatial autocorrelation is not very high. The CA parameters obtained by SAR regression are shown in Table 2. The spatial lag parameter *λ* is 0.9125 and the constant term *θ* is 16.3855. The influence of spatial variables from strong to weak is distance to water (*D_water_*, −16.5288), slope (*Slope*, 14.4775), population density (*Population*, 14.2627), GDP (*GDP*, 10.5513), distance to major roads (*D_roads_*, 10.2794), distance to railways (*D_railways_*, −7.1392), and distance to the center of the county (district) (*D_center_*, 2.7102). We can see that slope has become one of the main contributors, being different from many other cities, which can be attributed to the geographical conditions of Chengdu. The existing core construction area is located in a relatively gentle plain, while the fringe area of Chengdu is situated in a hilly region, on which it is difficult to build owing to the steep incline. According to the relevant regulations on grading technology of terrain and slope in the second national land survey, land with a slope exceeding 25° is not suitable for construction, which is in keeping with the result obtained from the statistical analysis of the slope of the existing built-up area in Chengdu. Therefore, for Chengdu, the slope is an important geographical factor that affects the urban spatial growth. The transition probability P_r affected by the spatial variables at 100 m resolution is obtained by applying the established transition rules. The overall transition probability map of the cells is made by multiplying P_r with the other terms in Equation (5), as shown in Figure 6. It can be seen from the figure that Pengzhou, Dayi, Qionglai, Pujiang, Longquanyi, and part of Dujiangyan have low transition probability, while that of Chongzhou, Xinjin, Wenjiang, Pidu, and part of Xindu and Jintang is high. Except for the built-up central town, the highest transition probability of Chengdu appears in Shuangliu and Dujiangyan, which also have predominant prospects for development.

### 4.2. Simulation Based on CA and Markov Chains

According to the principles of multi-criteria evaluation and dual-objective planning, the land-use change model is established on the basis of CA and Markov chains (Equations (1)–(3)). The proportion of spatial variables is adjusted, taking into account the spatial autocorrelation. Therefore, in this study, under natural development scenarios that are only based on their own development process without considering changes of factor, such as human activities and natural conditions, we take the land-use pattern in 2005 as the initial state and use the CA model to simulate and set the time interval between the simulation period and the base period as 10 years. Assign 0.0 to the background area in the output conditional probability map to keep the area as the background and set a permissible error ratio of 0.05 within the normal range (0.15). The larger the error ratio, the more likely it is to overestimate the amount of land-use change, thereby exacerbating the deviation of the simulation. The simulated land-use pattern in 2015 is obtained by comparing with the actual one in 2015 after calculation, as shown in Figure 4 and Figure 7. Tested by the kappa index, the accuracy of the built-up land spatial simulation is 0.7634, while the kappa index of non-built-up land spatial simulation is 0.9157, indicating that the simulation result of the model has a good accuracy and a certain reference value. Adopt the urban land-use pattern in 2015 as the initial state and use the constructed transition rules to get further simulations of the land use in 2025 and 2035, as shown in Figure 8 and Figure 9, respectively.

## 5. Discussions

A cellular automata (CA) model based on spatial autoregressive (SAR) and Markov chain is established, and the simulation of the urban spatial simulation of Chengdu is completed by combining with GIS technology. CA–Markov is a model widely used in similar research topics but it is difficult to effectively address the problem of the spatial autocorrelation of spatial variables. At the same time, compared with the FLUS [59] model and the CLUE-S [24] model widely used at present, the model constructed in this paper can not only simulate the spatial change of land use with higher precision but also dig out the influence degree of different spatial variable factors on urban spatial change, thereby providing a certain reference significance for the formulation of urban development planning. Thus, a SAR-based CA–Markov can not only express a more reasonable pattern of urban space, but it also promotes the formulation of urban development strategies by effectively considering spatial autocorrelation based on previous research. If we fail to reject the spatial autocorrelation hypothesis, it cannot be ruled out. In fact, we cannot ensure that this assumption would be rejected in other pilot areas, for different cities in China have different conditions, orientations, and development concepts. Based on the new-type urbanization pilot project and garden city construction, Chengdu has avoided the excessive density of space, compared with developed coastal areas. While the transition behaviors in this study are applicable to other scenarios, they should be combined with specific rules.

Moreover, the study results suggest that the fitting residuals are randomly distributed with very weak spatial autocorrelation characteristics and the SAR model is excellent in terms of the descriptive statistics of residuals and the spatial autocorrelation effect. The fitting parameters reveal that the effects of distance to water and major roads contribute much to the parameters; that is, the closer land is to water and major roads, the more likely it is to develop into built-up land. In addition, social effects such as population density and GDP are well-reflected in space, and land with greater population density and higher GDP tends to develop into built-up land more easily. According to the kappa index validation, the accuracy of space simulation of built-up land in 2015 is 0.7634, while that of the non-built-up land is 0.9157, which demonstrates the good accuracy and certain reference significance of the simulation results. After the further simulation, the simulation results of the land-use pattern of Chengdu in 2025 and 2035 are obtained, and the results are basically in line with the current development strategy of “advancing eastward, expanding southward, controlling westward, renovating northward and optimizing center” by reference to the Overall Urban Planning of Chengdu (2016–2035) and the Chengdu Chongqing Urban Agglomeration Development Plan (2016). Meanwhile, Jintang and Longquan, the regional centers on the development axis of Chongqing and Chengdu, would work as other engines for the economic and social development of Chengdu. The southern area would develop into an urban functional expansion zone, becoming the source of high-tech industries. The north–south city on the central axis of the new growth pole, together with the central town, constitutes the core functional area of Chengdu. The west is the functional area of a modern agricultural and ecological reserve consisting of the three satellite cities of Dujiangyan, Pidu, and Wenjiang and the five regional center cities of Pengzhou, Chongzhou, Dayi, Qionglai, and Pujiang, strictly limiting the development intensity of the west to 23%. The development of the northern industrial area focuses on speeding up the transformation of old industrial bases and communities and improving the basic public service functions and the livable environment. Consequently, it mainly focuses on internal optimization. In 2010, Chengdu implemented a series of functional dredgings of wholesale markets, warehouses, and industrial activities to guide the outward transfer of parts of the population and industries, thereby relieving the pressure of the central town. As a result, based on the spatial pattern, the simulation results are in line with the overall north–south development of Chengdu, and the simulation accuracy is great. In summary, the SAR–CA–Markov model can effectively obtain the main driving forces behind the rapid development of a garden city and determine their contributions. In future research, the integration of various techniques, theories, and models could help to obtain effective simulation ways for practical application.

## 6. Conclusions

This study develops a modeling framework for simulating the urban growth of Chengdu, a garden city, by integrating Markov chain and cellular automata with spatial autoregression. In terms of the model, the integrated model fully considers the spatial autocorrelation of various factors. It can also determine and select the appropriate driving factors according to the micro behavior pattern. Besides, it infers the possible situation of urban growth from the bottom up and analyzes the contribution of land policies that cannot be ignored in urban development. This framework combines Markov chain and CA with spatial autoregression, while inheriting their corresponding merits. Meanwhile, the objective of this framework is to simulate the pattern and form of land use that will appear soon considering spatial autocorrelation.

The results suggest that the development of the circle layer and the north–south axis in Chengdu are the consequences of adapting to and making full use of the physical geographical conditions, the development goals and requirements, and the historical stage of urban development. The results also imply that mastering the driving forces of urban development is the first step to improving land management. In addition to slope limitation, the closer land is to water and major roads, the more likely it is to develop into built-up land, and land with a higher population density and higher GDP is more likely to develop into built-up land. The simulated map and the actual image passed the kappa test after validation, for which the integrated model is an effective tool for simulating the development pattern and direction of a garden city. The integrated spatial autoregressive model was applied to a land-use planning practice in Chengdu, China, and obtained a series of results, which could effectively support the government and decision-makers in formulating appropriate policies for land-use planning in Chengdu, China. Furthermore, this case study has proven the effectiveness, superiority, and practicability of this model. Fully considering the development conditions and objectives of the research area and the difficulty of data collection, the model can be applied around the world.

Moreover, GIS technology, GeoDa software, and other tools were also used throughout the process. However, some research gaps in the discussion part still need to be bridged in future studies. Last, but not least, this framework has not been applied to other cities or regions, for which we plan to verify this framework by using it in more practical cases.

## Figures and Tables

**Figure 1 ijerph-19-11732-f001:**
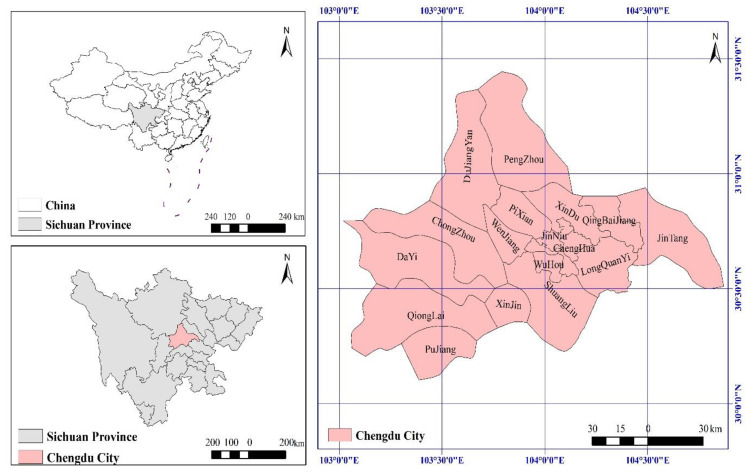
County map of Chengdu.

**Figure 2 ijerph-19-11732-f002:**
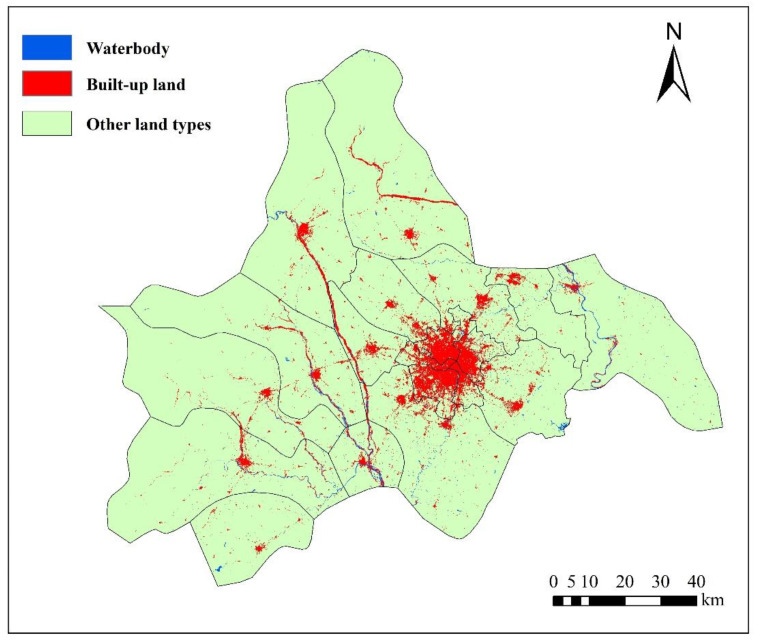
Land-use map of Chengdu in 2005.

**Figure 3 ijerph-19-11732-f003:**
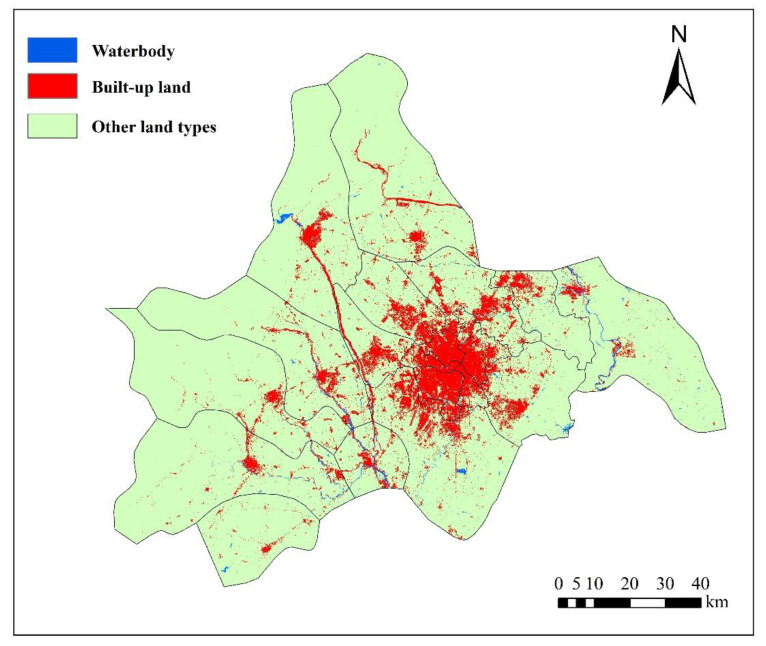
Land-use map of Chengdu in 2015.

**Figure 4 ijerph-19-11732-f004:**
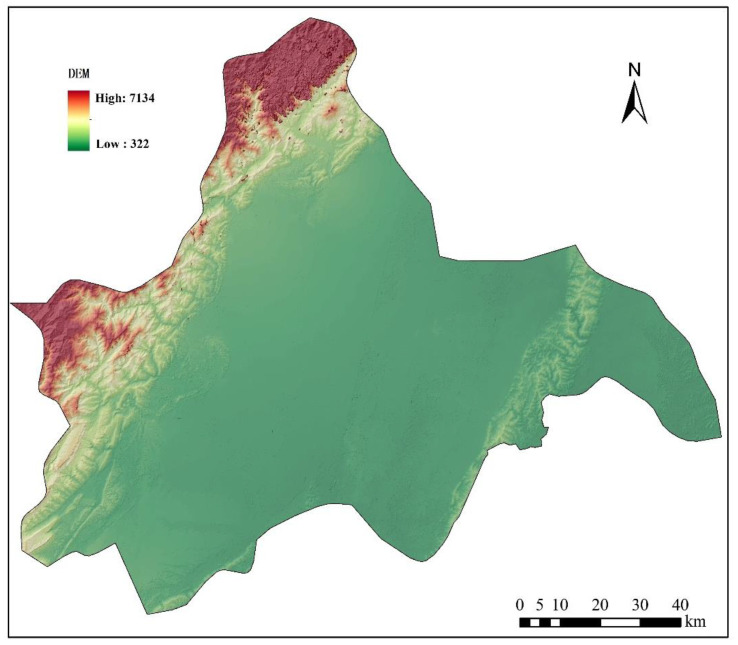
The elevation map of Chengdu.

**Figure 5 ijerph-19-11732-f005:**
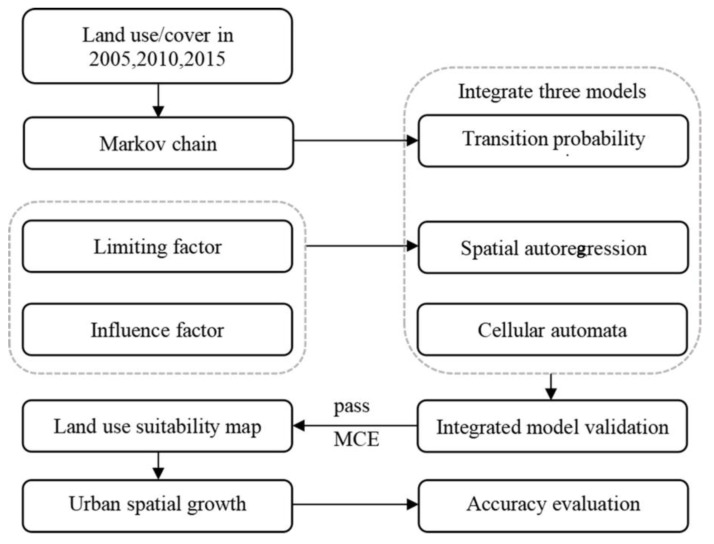
The framework of the optimized simulation process.

**Figure 6 ijerph-19-11732-f006:**
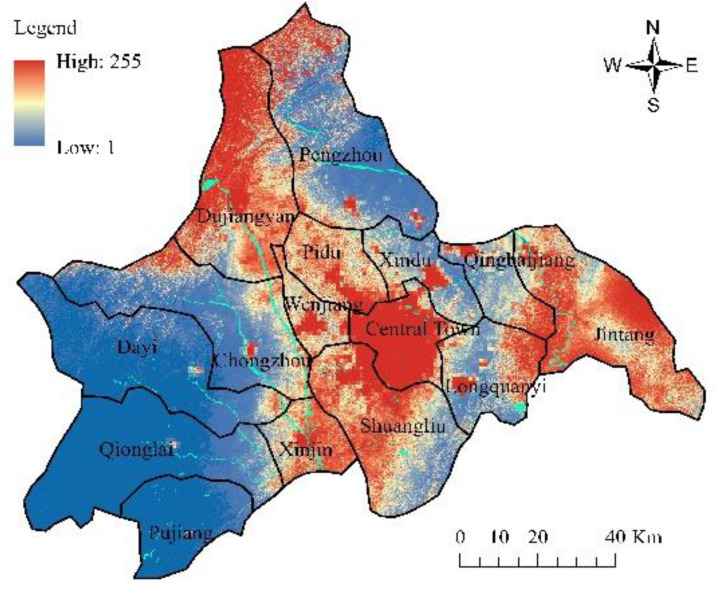
The land transition probability *p*.

**Figure 7 ijerph-19-11732-f007:**
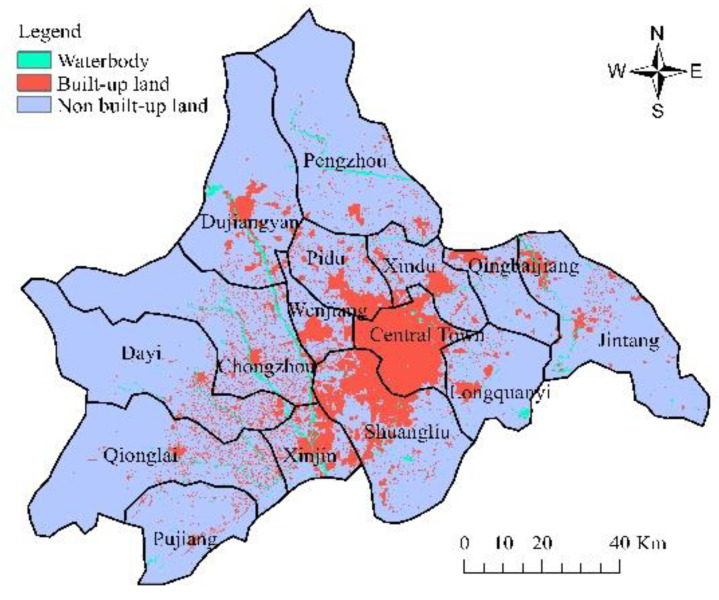
Simulation result of land use in 2015.

**Figure 8 ijerph-19-11732-f008:**
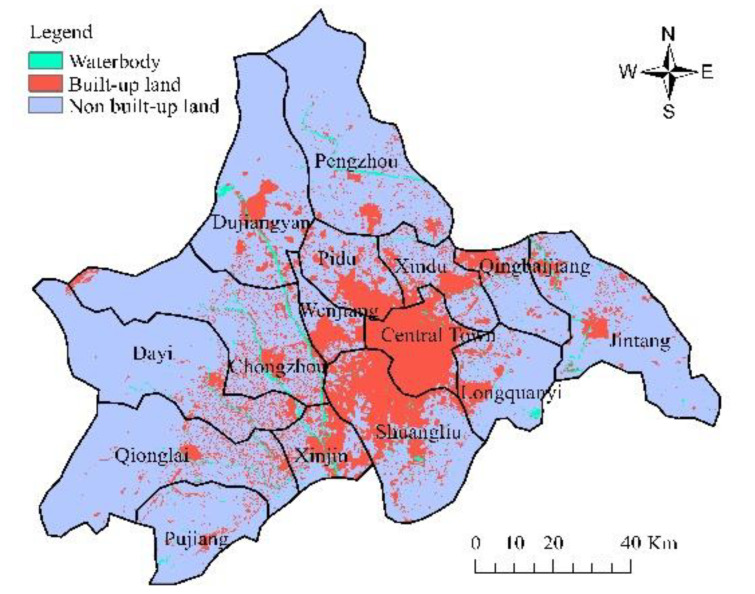
Simulation result of land use in 2025.

**Figure 9 ijerph-19-11732-f009:**
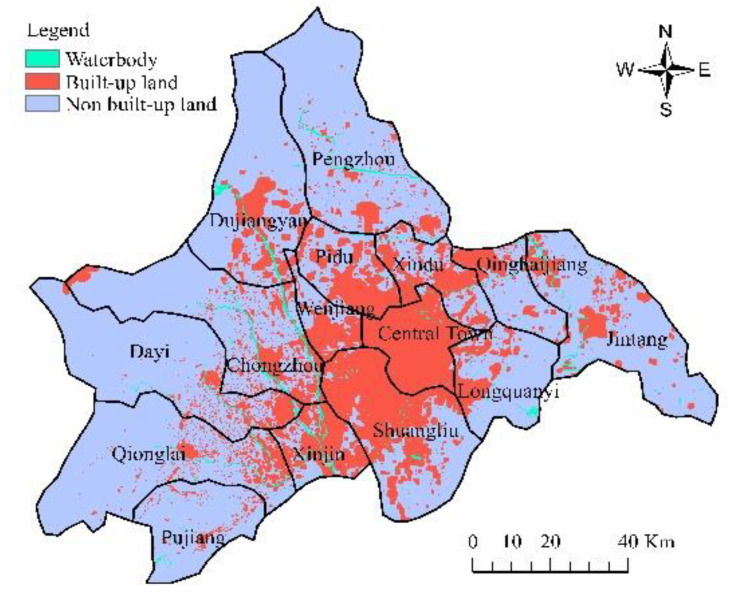
Simulation result of land use in 2035.

**Table 1 ijerph-19-11732-t001:** Fitting performance for spatial variables.

Model	Residuals’ Descriptive Statistics		Residuals’ Space Autocorrelation	
	Average Value	Sum of Squares	Moran’s I	*p*-Value
SAR	−4 × 10^−8^	2797.8751	0.0225	0.3087

**Table 2 ijerph-19-11732-t002:** The CA parameters obtained by SAR.

λ	θ	*D_center_*	*D_roads_*	*D_railways_*	*D_water_*	*GDP*	*Population*	*Slope*
0.9125	16.3855	2.7102	10.2794	−7.1392	−16.5288	10.5513	14.2627	14.4775

## Data Availability

All data generated or analyzed during this study are included in this article.

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
