# Peer review of "An Integrated Spatial Autoregressive Model for Analyzing and Simulating Urban Spatial Growth in a Garden City, China"

_ijerph, 2022, doi:10.3390/ijerph191811732_

Round 1

Reviewer 1 Report (Previous Reviewer 3)

30 / 5 000  

Wyniki tłumaczenia

Thank you for the corrections. Best regrads

Author Response

Thank you very much for your comments and suggestions. We will continue to check the full text and further improve the quality of the manuscript. Thank you again for your review.

Reviewer 2 Report (New Reviewer)

The study presents a modelling method of urban growth stimulation, which integrates Markov chain and Celluar Automata with spastial autoregression. Authors used Chengdu as an empirical case to test this stimulating model.

First, as the topic of stimulating urban growth has been widely discussed in literature, authors are strongly suggested to present clearly how this study can contribute to the existing knowledge. 

Second, the authors should revise the writing logic and phraseology to improve consistency and readability. For example,  lines 285-286 suggest the aims of this study, while the following passages do not belong to the objectives of the whole study. 

Third, the association of this integrated model with garden city should be clarified. Why this model is more suitable for stimulating the urban spatial growth for a garden city? Is it applicable for other types of cities?

Author Response

The study presents a modelling method of urban growth stimulation, which integrates Markov chain and Celluar Automata with spastial autoregression. Authors used Chengdu as an empirical case to test this stimulating model.

First, as the topic of stimulating urban growth has been widely discussed in literature, authors are strongly suggested to present clearly how this study can contribute to the existing knowledge.

Response: Thank you very much for your suggestions. We explained it in the revised lines 158-166 and 436-452.

Second, the authors should revise the writing logic and phraseology to improve consistency and readability. For example, lines 285-286 suggest the aims of this study, while the following passages do not belong to the objectives of the whole study.

Response: Thank you very much for your suggestions. According to your suggestion, we modified the expressions in 285-286. Please see the revised lines 288-303.

Third, the association of this integrated model with garden city should be clarified. Why this model is more suitable for stimulating the urban spatial growth for a garden city? Is it applicable for other types of cities?

Response: Thank you very much for your suggestions. The integration model we designed is illustrated by taking the garden city as an example to explore its driving factors of urban expansion. The method used is not limited to this type of city, which requires the reader to choose according to the situation of the study area.

This manuscript is a resubmission of an earlier submission. The following is a list of the peer review reports and author responses from that submission.

Round 1

Reviewer 1 Report

This study builds an integrated model for the analysis and simulation of urban growth. The authors have made a lot of efforts on the extensive literature review, but there are a couple of issues to be consideration for the publication in the International Journal of Environmental Research and Public Health.

Major Comments.

  1. I think the authors may wish to restructure the paper. The current from of the introduction includes the problem statement and the literature review. It is relatively long to have the extensive review in the introduction. So, it would be better to have the section called literature view.

  1. The author does not have to review all methods related to the urban simulation model. I think only related methods (CA, ABM, authoregressive model) may need to be contained. I think SD seems not to be relative.

  1. The framework should be a part of the method, which should not be included in the introduction.
  2. Just simply describing the general concept of the method is not methodology. It should be described with how the method was used in the model. The author needs to be more tailored.

  1. In general, the study area comes first and then method appears.

  1. The author should improve the quality of the maps.

  1. There is no any details of the simulation scenario or something. Without any explanations, the results may not help the readers to understand the paper.

  1. The author mentioned the main contribution of this paper is to model the framework for simulation. Such developing the model itself should not be the contribution. I think the authors need to rethink the research questions (or research objectives).

Minor Comments

  1. Wrong citation (line 285)
  2. There are some words without explanation (e.g., A universal approach, urban modeling, etc). Those words need to be explained.

Author Response

RESPONSES TO THE REVIEWER 1’S COMMENTS

We are grateful to the Reviewer 1 for his/her insightful review. His/Her comments have contributed substantially to improving the paper. According to he/she, we have made efforts to significantly revise the manuscript, with the details explained as follows.

Major Comments.

  1. I think the authors may wish to restructure the paper. The current from of the introduction includes the problem statement and the literature review. It is relatively long to have the extensive review in the introduction. So, it would be better to have the section called literature view.

Response: Thank you for your comments. In the revised version, we have restructured the paper and divide the introduction into two parts: problem statement and literature viewpoint.

  1. The author does not have to review all methods related to the urban simulation model. I think only related methods (CA, ABM, authoregressive model) may need to be contained. I think SD seems not to be relative.

Response: Thank you for your comments. In the revised version, we have removed the literature review of the irrelevant models.

  1. The framework should be a part of the method, which should not be included in the introduction.

Response: Thank you for your comments. In the revised version, we moved the framework of the introduction to the methods section.

  1. Just simply describing the general concept of the method is not methodology. It should be described with how the method was used in the model. The author needs to be more tailored.

Response: Thank you for your comments. In the revised version, we have added a description of how to use the method in the model.

  1. In general, the study area comes first and then method appears.

Response: Thank you for your comments. In the revised revision, we have moved the research area prior to the Methods section.

  1. The author should improve the quality of the maps.

Response: Thank you for your comments. We have improved the quality of the maps.

  1. There is no any details of the simulation scenario or something. Without any explanations, the results may not help the readers to understand the paper.

Response: Thank you for your comments.  We have added a description of the simulation scenario.

  1. The author mentioned the main contribution of this paper is to model the framework for simulation. Such developing the model itself should not be the contribution. I think the authors need to rethink the research questions (or research objectives).

Response: Thank you for your comments. We have reworked the contribution section. The model should not be the main contribution of the paper, the contribution of this paper should be to predict the development trend of land use in Chengdu city by using the combined model, and the results effectively support the government and policy makers in formulating land use planning in Chengdu city, etc.

Minor Comments

  1. Wrong citation (line 285)

Response: Thank you for your comments. We have added the correct citation. . Please see this revised text in lines 193-195.

  1. There are some words without explanation (e.g., A universal approach, urban modeling, etc). Those words need to be explained.

Response: Thank you for your comments. We have added some words to explain . Please see this revised text in lines 50-54.

Generally, we are deeply grateful to the reviewer 1 for his/her insight and careful review. His/her comments have greatly helped improve the paper. We also expressed our gratitude in the "Acknowledgment" section of the revised manuscript.

Reviewer 2 Report

In this manuscript, the authors perform a study constructs a model consisting of Spatial AutoRegressive (SAR), Cellular Automata (CA), and Markov chain. It can consider spatial autocorrelation well, and integrate the geographical advantages of CA simulation, to help seek the driving forces behind of garden city expansion. However, I will comment on some aspects to improve this article:
-The format of the bibliographic citations is incorrect, the authors must review the format of the MDPI.
-The acronyms are spelled incorrectly. The correct form to write them is with their capital letters meaning the acronym, such as "Cellar Automata (CA)". This error must be fixed in all acronyms that have been written in the manuscript.
-Do not place references with hyperlinks.
-In the Introduction Section, it must be separated into two Sections, the first one for Introduction and the second one for Related Works.
-At the end of the Introduction Section or Related Works, the authors must write a brief introduction of each Section that will come later.
-Improve the titles of the Sections and Subsections.
-Authors must not write with Phrasal Verbs in a scientific article.
-Matrices and Equations are off the edge of the manuscript.
-The authors must write a brief introduction after each Section or Subsection.
-Sections are not in the corresponding verb tense.
-The authors do not reference the Equations in the text.
-Some Figures are far from the text where they are referenced.
-Figure 3 to 9 do not have axes of latitude and longitude.
-References do not correspond to the MDPI format.
-There are very old bibliographical references, it is recommended references no more than 10 years old.

Author Response

RESPONSES TO THE REVIEWER 2’S COMMENTS

We are grateful to the Reviewer 2 for his/her insightful review. His/Her comments have contributed substantially to improving the paper. According to he/she, we have made efforts to significantly revise the manuscript, with the details explained as follows.

  1. In this manuscript, the authors perform a study constructs a model consisting of Spatial AutoRegressive (SAR), Cellular Automata (CA), and Markov chain. It can consider spatial autocorrelation well, and integrate the geographical advantages of CA simulation, to help seek the driving forces behind of garden city expansion. However, I will comment on some aspects to improve this article:

  1. The format of the bibliographic citations is incorrect, the authors must review the format of the MDPI.

Response: Thank you for your comments. We have revised the format of the bibliographic citations

  1. The acronyms are spelled incorrectly. The correct form to write them is with their capital letters meaning the acronym, such as "Cellar Automata (CA)". This error must be fixed in all acronyms that have been written in the manuscript.

Response: Thank you for your comments. The problem of incorrect acronyms is fixed. All the terms in this paper, such as cellar automata, we have addressed them with capital letters, such as Cellular Automata.

  1. Do not place references with hyperlinks.

Response: According to your opinions, we have addressed the problem that references are hyperlinks. Now all the references are cross-references, not hyperlinks, and you can click each of them to go to the location in which the reference is.

  1. In the Introduction Section, it must be separated into two Sections, the first one for Introduction and the second one for Related Works.

Response: Thank you for your comments. The introductory part of this article is already divided into two parts, the last paragraph being the relevant work and implications.

  1. At the end of the Introduction Section or Related Works, the authors must write a brief introduction of each Section that will come later.

Response: Thank you for your comments. We have added a brief description of each section that will come later at the end of the related work.

  1. Improve the titles of the Sections and Subsections.

Response: Thank you for your comments. We have improved the titles of some of the subsections.

  1. Authors must not write with Phrasal Verbs in a scientific article.

Response: Thank you for your comments. We have replaced some of the phrasal verbs in the article.

  1. Matrices and Equations are off the edge of the manuscript.

Response: Thank you for your comments. This problem you mentioned we have addressed. All the matrices and equations are at the middle of the paper, and all the numbers of them are at the right edge of the paper.

  1. The authors must write a brief introduction after each Section or Subsection.

Response: Thank you for your comments. We have written a brief introduction after each subsection where it is necessary.

  1. Sections are not in the corresponding verb tense.

Response: Thanks for your comments. The problem of incorrect verb tense has been fixed.

  1. The authors do not reference the Equations in the text.

Response: Thank you for your comments. We have referred to the equations where necessary in the text.

  1. Some Figures are far from the text where they are referenced.

Response: Thank you for your comments. For ease of viewing and comparison, we have placed the figures after the relevant text in each subsection.

  1. Figure 3 to 9 do not have axes of latitude and longitude.

Response: Thanks for your comments. We have revised the figures.

  1. References do not correspond to the MDPI format.

Response: Thanks for your comments. We have revised the format of references.

  1. There are very old bibliographical references, it is recommended references no more than 10 years old.

Response: Thanks for your comments. The problem you mentioned that some references are too old we have addressed. We renew all the references and delete the old ones.

Generally, we are deeply grateful to the reviewer 2 for his/her insight and careful review. His/her comments have greatly helped improve the paper. We also expressed our gratitude in the "Acknowledgment" section of the revised manuscript.

Reviewer 3 Report

Dear Authors

There is no verification of the conducted research in the text, I believe that it would be important either to refer to the spatial changes that occurred in the area in question in the time interval, eg 5 and 10 years in the past. Or as a verification, some other example should be discussed in which the discussed method was applicable and brought specific results.

Best regards

Reviewer

Author Response

RESPONSES TO THE REVIEWER 3’S COMMENTS

We are grateful to the Reviewer 3 for his/her insightful review. His/Her comments have contributed substantially to improving the paper. According to he/she, we have made efforts to significantly revise the manuscript, with the details explained as follows.

There is no verification of the conducted research in the text, I believe that it would be important either to refer to the spatial changes that occurred in the area in question in the time interval, eg 5 and 10 years in the past. Or as a verification, some other example should be discussed in which the discussed method was applicable and brought specific results.

Response: Thank you very much for your suggestion. We just use one city to verify the proposed framework, we will continue to verify the modelling practicability in the future research; However, in this study we have verified the modelling accuracy use Kappa coefficient (76%) which is conformed to model requirements. We added a supplement in the conclusion section to explain the future research direction. Furthermore, We have revised the analysis and discussion of the results in this paper to emphasize the specific results.

Generally, we are deeply grateful to the reviewer 3 for his/her insight and careful review. His/her comments have greatly helped improve the paper. We also expressed our gratitude in the "Acknowledgment" section of the revised manuscript.

Round 2

Reviewer 2 Report

Thanks to the authors for making the suggested changes. Before publishing the manuscript, authors should correct the references and improve the language and presentation of the manuscript because they have strikethrough and it does not represent good quality.

Author Response

Thank you very much for your meanful comments. In the revised version 2, we have done our best to correct the references and improve language and presentation.